# CopyLens: Dynamically Flagging Copyrighted Sub-Dataset Contributions to LLM Outputs

## Abstract

Large Language Models (LLMs) have become pervasive due to their knowledge absorption and text-generation capabilities. Concurrently, the copyright issue for pretraining datasets has been a pressing concern, particularly when generation includes specific styles. Previous methods either focus on the defense of identical copyrighted outputs or find interpretability by individual tokens with computational burdens. However, the gap between them exists, where direct assessments of how dataset contributions impact LLM outputs are missing. Once the model providers ensure copyright protection for data holders, a more mature LLM community can be established. To address these limitations, we introduce *CopyLens*, a new framework to analyze how copyrighted datasets may influence LLM responses. Specifically, a two-stage approach is employed: First, based on the uniqueness of pretraining data in the embedding space, token representations are initially fused for potential copyrighted texts, followed by a lightweight LSTM-based network to analyze dataset contributions. With such a prior, a contrastive-learning-based non-copyright OOD detector is designed. Our framework can dynamically face different situations and bridge the gap between current copyright detection methods. Experiments show that *CopyLens* improves efficiency and accuracy by 15.2% over our proposed baseline, 58.7% over prompt engineering methods, and 0.21 AUC over OOD detection baselines.

## 1 Introduction

Following the release of GPT-3 (Brown et al. (2020)), LLM-driven applications have gained increasing attention in both industry and academia. As transformer architectures mature, the demand for high-quality data has become urgent. However, a major security concern that hinders the establishment of the LLM community is the challenging task of protecting dataset copyright. Specifically, the current model providers lack the capability to identify which datasets contribute most to a response. This inability makes data holders hesitant to expose their datasets to LLM training since copyrighted datasets are of high quality but no credit is given to themselves. Conversely, if model providers could pinpoint the contributions of datasets to a response, a beneficial transactional framework between the data holder and LLM customer could be established. (Cui & Araujo (2024))

**In order to build a complete copyright dataset contribution analysis framework, two stages are needed. First, the framework detects whether LLM outputs involve copyrighted contents, then analyzes the sub-dataset contributions to outputs identified as copyrighted.** However, most works either focus only on the first stage with fixed tasks, i.e. only detecting verbatim sentences from books, or only on finding explainability in tokens, which is similar to the second stage but needs huge computation. (Duarte et al. (2024); Shi et al. (2023); Vyas et al. (2023)) Besides detecting identical copyrighted contents, previous studies tackle copyright issues from three aspects: erasing copyrighted data, protecting pretraining data, and assessing data impact. For data erasure, the need for finetuning in the LLM unlearning approach is computationally heavy and unreliable. (Eldan & Russinovich (2023); Yao et al. (2023); Min et al.; Shi et al. (2023)). In pretraining data protection, watermarking for source traceability degrades model performance and loses trace accuracy in generated texts.(Wu et al. (2023)) Therefore, we mainly focus on assessing data impact.

Data impact assessment generally includes mathematical methods and prompt engineering ones. Despite works to build similarity metrics or gradient-based explanations, most mathematical methods

are preliminary, focusing on conceptual proof with complex mathematical functions. (Scheffler et al. (2022); Grosse et al. (2023); Deng et al. (2024)) Prompt engineering in LLM attribution aims to ask "what the response is based on", emphasizing factual grounding in cited documents rather than identifying styles or sub-datasets contributions. This approach differs in goals and may amplify hallucinations. (Zuccon et al. (2023)) In summary, detecting verbatim itself cannot avoid copyright infringement with particular elements, while mathematical methods take up too much computation. The gap between such a fixed task and unreasonable computational cost exists in data impact assessment, blocking the evaluation of which datasets contribute most to LLM outputs.

In this work, we focus on identifying the pretraining datasets that contribute most significantly to LLM outputs, and corresponding efficient non-copyright detectors. Current works target copyright by finding statistical differences in loss or final layer logits outputs to distinguish copyright verbatims. **Can more information be revealed by looking into LLM's inherent architecture from the model provider's perspective?** Inspired by this, our framework explores the information within LLM's hidden states. However, the development of such a framework presents several challenges. Firstly, which inherent information affects effective attribution in this task is unknown. Secondly, the difference between non-copyright detection and contribution analysis of copyrighted ones makes it hard to combine them into a single framework. Thirdly, due to the large size of LLMs, the copyright analysis framework must be lightweight and efficient.

To protect dataset copyright in LLMs, we propose *CopyLens*, a framework that both extracts representations from LLM dialogues to calculate the contribution of training datasets, and detects non-copyright LLM outputs. We address the unclear problem of copyright analysis by first abstracting it into a sub-dataset source classification problem, then extending to more realistic generation scenes to prove effectiveness. Our framework leverages patterns in Multi-Head Attention (MHA) where LLM outputs are linked to datasets. By exploring temporal dependency in layer-wise MHA, we build a lightweight LSTM-based framework that efficiently performs dataset-level copyright analysis. Based on such LSTM temporal prior, an efficient built-in non-copyright detector is proposed. We validate our framework on BERT, GPT series, demonstrating comparative performance with limited training samples while applicable to different generation scenes. Our contributions can be summarized as follows:

- Recognizing the need for dataset-level copyright in the LLM community, we introduce *CopyLens*, a lightweight, LLM-training free framework that can both identify the contribution among different training datasets to LLM outputs and detect non-copyright ones with limited training samples.

- We emphasize the necessity of copyrighted data information triggered in LLM architecture by looking into hidden space. Based on three effective information extraction methods, an LSTM-based copyright analysis framework that aggregates the inherent layer-wise MHA relationship in LLMs is proposed. *CopyLens* shows classification accuracy over 94.9% in less than 1.5 hours training time.

- An efficient non-copyright OOD detector is designed based on LSTM prior information with 93.83% accuracy and an AUC score of 0.954, which surpasses previous OOD detection methods. Considering several generative types that involve copyright issues, we design prompts and employ the generated LLM outputs in different scenes to evaluate our framework. For the qualitative study, *CopyLens* aligns with baselines, showing better detection in three controlled scenes. Quantitatively, *CopyLens* shows an average MSE loss under 0.05 in the text mixture task.

## 2 BACKGROUND

Driven by extensive demand for refining texts, LLM-assisted editing offers substantial potential. A critical issue is copyright protection, particularly when models use multiple copyrighted sub-datasets. Specifically, it is important to determine whether the generated LLM outputs is based on such datasets, and if so, which ones matter? We first outline the copyright challenges inadequately addressed before, review existing methods, then overview the architecture basic of LLM.

## 2.1 Copyright Takedown and Training Data Assessment on LLM Outputs

Early LLMs gathered datasets through web crawling, which limits manual review of copyrighted material. Patterns of copyrighted works may be replicated, leading to resembled generation. A brief overview of the most relevant research is provided here, with an extended one in the Appx. A.3.

**Copyright Takedown in LLMs** Copyright takedown starts by detecting related content. As LLM shows properties different from small models when scaling, such as finetuned loss, output logit distribution, paraphrasing attack, and perplexities, such features help differentiate copyrighted texts. (Li et al. (2024); Shi et al. (2023); Duarte et al. (2024); Li (2023) ) However, simply tracking LLM outputs or prompting cannot deal with complex tasks. Despite recent evaluation, the general copyright analysis framework is not yet defined, with token length limited to 200, and small datasets limited to books and news. Therefore, large improvement space remains. (Wei et al. (2024))

**Mathematical and Prompt Engineering Training Data Assessment** How does the inclusion of copyrighted material as training data in open-source models affect the resulting outputs? Mathematical approaches like K-Near Access-Free (K-NAF) similarity have been developed, but these methods are preliminary and not universally applicable (Vyas et al. (2023); Scheffler et al. (2022); Elkin-Koren et al. (2023)). The mathematical mechanisms of such problems are not clear, making attempts limited even if token and layer-level analysis is studied. (Wang et al. (2023b)) Despite current efforts in making gradient-based influence functions computable, computing gradients for all training data is still an unacceptable cost. (Grosse et al. (2023))

Prompt engineering is currently used in LLM attribution, asking "What is the answer based on". Direct attribution refers to asking the LLMs to provide attribution while answering. (Sun et al.) Post-retrieval means answering and then asking the LLMs to attribute based on it, while post-generation attribution uses both questions and answers. (Reddy et al. (2023); Gao et al. (2022)) However, such works mainly focus on whether the outputs are facts grounded on pretraining data. Besides, it shows hallucinations and knowledge conflicts in professional fields. (Zuccon et al. (2023); Li et al. (2023))

## 2.2 Architecture Basic in LLMs

**LLM Architecture** Despite efforts in copyright dataset protection, few look into it from the model provider's perspective, neglecting the importance of the generation process in LLM architecture.

The architecture of LLMs comes from Feed-Forward Networks (FFN) and MHA in transformers. By projecting the query, key, and value, MHA aggregates tokens by calculating attention scores, which allows to focus on different segments in separated representation subspaces.

For single head attention, let $X \in \mathbb{R}^{n \times d}$ denote the embeddings of all tokens, $W^K, W^Q, W^V \in \mathbb{R}^{d \times d_h}$, where $X$ and $Y$ stand for the input and output of one head in an MHA layer, and $d$ represents the maximum token length supported by an LLM.

$$Q = XW^q, K = XW^k, V = XW^v, \tag{1}$$

$$Y = softmax(\frac{QK^T}{\sqrt{d}})V \tag{2}$$

**Causality in MHA** Attention has been studied for causality recently, shifting from using attention weights for input-output relations explanation, to estimating symbolic structural relationships in pre-trained LLMs. (Rohekar et al. (2024); Nisimov et al. (2022))

## 3 Implementation

Firstly, we emphasize the necessity of copyright protection for training datasets. Then, we formulate our copyrighted dataset contribution analysis problem. Lastly, we present our LSTM-based contribution analysis with information extraction strategies, and the non-copyright detector built on it.

## 3.1 Training Datasets' Copyright

In this work, we aim to build a more mature ecosystem around the LLM training dataset. Here we present the framework of an LLM-based application, the targeted two-stage problem, and our overall solution for the copyrighted training dataset protection.

### 3.1.1 LLM-BASED COMMUNITY AND COMMERCIAL BEHAVIORS

In the LLM-based application shown in Fig. 1, three participants are involved: a customer, a model provider, and multiple data holders. Initially, data holders supply the training datasets to the model provider to train a multifunctional LLM. Then, the customer can query the model provider and get the corresponding LLM response. Therefore, the customer should pay for the service once receiving the response. However, unlike a general point-to-point transaction, the customer should not only pay for the model provider but also pay for the data holders. The reason is that both the LLM and training datasets contribute to the responses received by the customer. Usually, the model provider can ask the customer to pay for different services. However, the challenge of bridging the gap between data holders and customers remains unaddressed. Thus, in this work, we propose *CopyLens*, a framework that can track which training datasets contribute most to a specific response received by the customer.

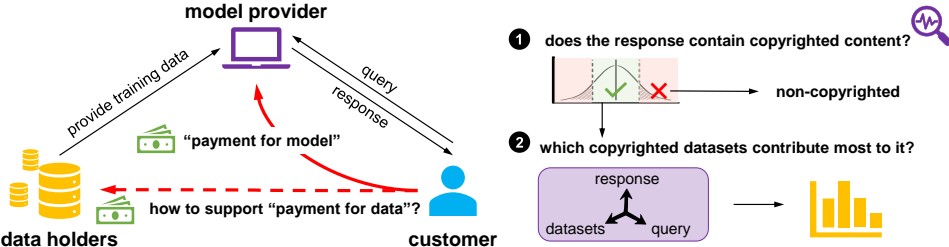

Figure 1: LLM-based community and commercial behaviors. Both copyright detection and dataset contribution analysis are key to supporting "payment for data" in this transactional framework.

### 3.1.2 PROBLEM FORMULATION

The problem is in two stages, which first need filtering out non-copyrighted LLM responses, then analyzing the contribution of copyrighted datasets to them, as depicted in Fig. 1. We adapt the notations from Hu et al. (2024), and define the problem as follows: Given a user-defined text input $T$, an LLM $M$ which is trained on the complete dataset $D$, which consist of both in-distribution (ID) and out-of-distribution (OOD) datasets, representing copyrighted and non-copyrighted data respectively. As shown in Eq. 3, where $D_{\text{ID}}$ denotes the set of copyrighted sub-datasets, and $D_{\text{OOD}}$ denotes non-copyrighted ones. The LLM generates response $R$ to each user input.

$$D_{\text{OOD}} = D \setminus D_{\text{ID}},$$
$$D_{\text{ID}} = \bigcup_{i=1}^{n} D_{\text{ID}}^i, \quad D_{\text{OOD}} = \bigcup_{j=1}^{m} D_{\text{OOD}}^j \tag{3}$$

**Stage 1: Filtering out non-copyrighted responses** This can be formulated as an OOD problem, as described in Eq. 4. The confidence score function takes in model $M$ and response $R$ for consideration to quantify how close $R$ resembles ID data. The non-copyright texts are filtered by a threshold value $\delta$, which is returned by decision function $I_\delta$.

$$I_\delta(M, R) = \begin{cases} R \in \text{Copyrighted} & \text{Confidence}(M, R) \geq \delta \\ R \in \text{Non-copyrighted} & \text{Confidence}(M, R) < \delta \end{cases} \tag{4}$$

**Stage 2: Copyrighted datasets contribution analysis** To get a soft contribution score $\mathbf{S_D} \in \mathbb{R}^n$ from copyrighted sub-datasets $D_{\text{ID}}$, parameterized function mapping $F_{\theta_{opt}} : R \mapsto \mathbf{S_D}$ is needed. Specifically, $\mathbf{S_D}$ denotes scores of each potentially contributing dataset in $D_{\text{ID}}$. Such contribution analysis is difficult because the generation varies with uncontrolled user inputs. To optimize $F_{\theta_{opt}}$ in this problem, we first reduce it to a simplified version, and then generalize it back to the original one. In the training stage, for each potentially copyrighted sub-dataset $k$, we optimize mapping strategy $F$ using the original datasets as input in a supervised way, as shown in Eq. 5. $\mathbf{I}_n \in \mathbb{R}^{n \times n}$ denotes the identity matrix since inputs are supervised, and we denote optimized parameters as $\theta_{sim}$:

$$\max_{\theta_{\text{sim}}} \sum_{k=1}^{n} P\left(F_{\theta_{\text{sim}}} : T \mapsto \mathbf{I}_{n,k} \mid M, T \in D_{\text{ID},k}\right) \tag{5}$$

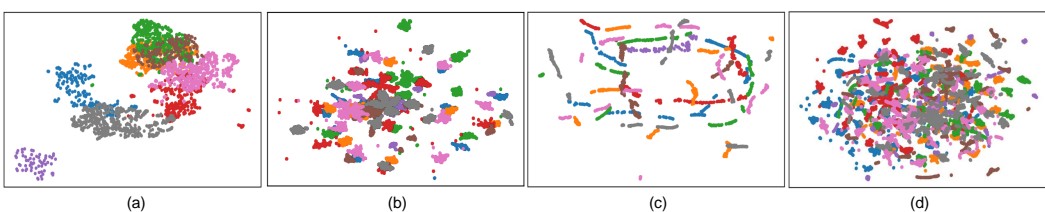

Figure 2: Visualization of the statistical differences in MHA outputs using UMAP with multi-class dataset input. (a,b) Outputs from an LLM (BERT) not trained on the dataset at layer 3 and across all layers, exhibiting distinct clustering behavior. (c,d) Outputs from an LLM (GPT-series) pre-trained on the dataset at layer 3 and across all layers, showing filamentous distribution.

Using this as a basic, we then extend to generalized scenarios and show similarities between the generalized ones and the original problem with optimized parameters $\theta_{opt}$.

$$P(F_{\theta_{sim}} : \boldsymbol{R} \mapsto \mathbf{S}_D) \simeq P(F_{\theta_{opt}} : \boldsymbol{R} \mapsto \mathbf{S}_D) \tag{6}$$

### 3.1.3 Two-Stage Solution Framework for Copyrighted Dataset Protection

**Different from previous methods that only focus on copyright dataset detection (stage 1), we combine two stages by using learned prior from contribution analysis of copyrighted datasets to LLM outputs (stage 2) to help build an efficient detector in stage 1.** The reason is that detailed copyright analysis in stage 2 allows us to learn rich semantics, which we leverage as prior knowledge to enhance efficiency and reduce redundancy in stage 1. Therefore, by first training stage 2 and fine-tuning embedding on stage 1, we propose an integrated two-stage solution framework.

The prior learning starts in stage 2. To process texts containing copyrighted elements, we first obtain layer-wise MHA outputs, inspired by Fig. 2, which will be discussed in the next section. To reduce computational costs while preserving key information, we propose three extraction strategies. After extracting representations, we apply an LSTM-based classifier with a softmax output layer to evaluate the contributions of potential datasets, as shown in Fig. 3(b). With the learned LSTM prior, in stage 1, we introduce a contrastive-learning-based non-copyright detector. This detector fine-tunes both the global projector and the pre-trained LSTM weights, as illustrated in Fig. 3(a).

### 3.2 Stage 2: Information Extraction and LSTM-based Contribution Analysis

**Layer-Wise Importance of Pretraining Data in MHA Embedding Space** Two trials have been conducted to analyze pretraining copyrighted text. First, using the BERT backbone as a feature extractor gains limited performance in more complex datasets, which fails to meet our expectations. Therefore, similar to analyzing output logits to find statistical differences in pretraining data, an intuitive idea is to investigate the last-layer MHA output of LLM. However, the performance gain is still limited, meaning different functionality across layers. To understand the differences, we adopt UMAP, a typical dimensionality reduction method that uses graph layout algorithms to arrange data into low-dimensional space. (McInnes & Healy (2018)) As illustrated in Fig. 2 (c), specific layers show a filamentous distribution in the MHA embeddings with pretraining data inputs, while in Fig. 2 (a), the untrained result does not. This indicates richer information than mere clustering, motivating us to incorporate layer-wise dependencies into our framework.

**Information Extraction Strategies** We use the MHA output as the extracted representation for an LLM dialog, which is then input into a copyright identification framework. However, the token length of $Y$ in Eq. 2 varies, and a consistent and significant input pattern is more friendly to a copyright identification framework. Since our goal is to develop an inference-only method for arbitrary dialog length, adapting LLM model architecture is not suitable because of additional fine-tuning. To this end, noticed by the sparsity of attention in the pre-trained language model, we design two main information extraction strategies (Interval-based & Variance-based, Fig. 8). These strategies leverage the capabilities of Multi-head Attention and are designed for future research inspiration.

**(1) $k$ token interval sampling** In this method, we directly sample $k$ tokens with fixed patterns in a dialog, as shown in Fig. 8. Specifically, when $k = 1$, the middle token is sampled in a dialog.

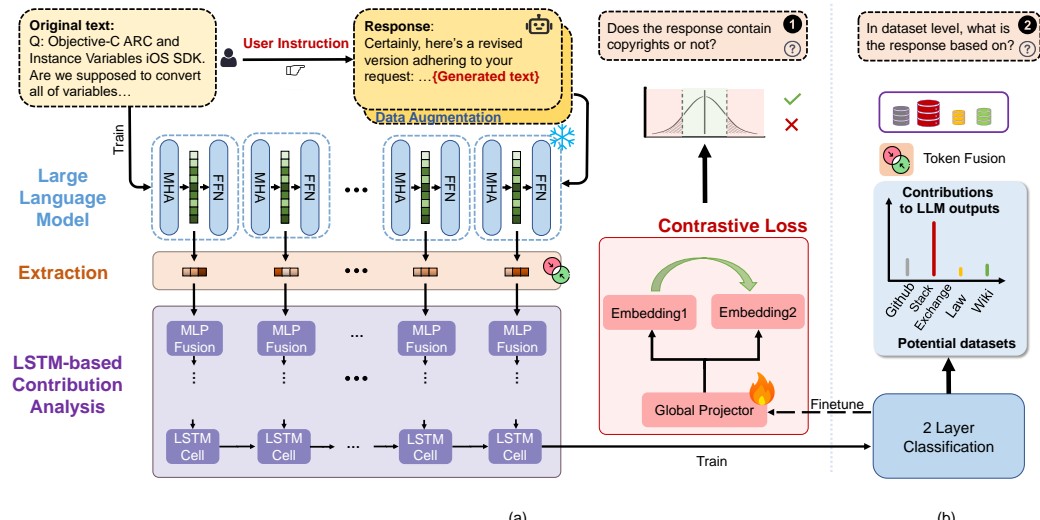

(a)

(b)

Figure 3: Two-stage dataset-level copyright analysis framework. (a) Stage 1: Detection of copyrighted texts. Using the learned LSTM prior from stage 2, first-layer MLP outputs in the classification module are fed into and fine-tuned with contrastive learning by the global projector, creating an embedding space that effectively distinguishes non-copyright texts by a defined threshold. (b) Stage 2: Dataset contribution analysis for LLM response with copyright issues. During inference, response text is fed into LLM, and representations from intermediate MHA output layers are extracted. Then they are fused to an LSTM-based classifier that returns each potential dataset's contribution score.

Otherwise, tokens with interval $n/(k-1)$ are sampled, where $n$ stands for the total number of tokens in a dialog. The procedure can be summarized as

$$\overline{S}_l = \begin{cases} \{Y_l[\frac{n}{2}]\} & k=1 \\ \{Y_l[i] \mid i = 0, \frac{n}{k-1}, \frac{2n}{k-1}...\} & k>1 \end{cases} \tag{7}$$

Here, we use $\overline{S}_l$ to represent the selected tokens in the $l^{th}$ MHA block. This sampling method is intuitive but effective at handling long-range context. For simplicity, we denote this method as **INTERVAL (INTER)** .

**(2) Top-K pivotal tokens** As shown in Fig. 8, for each token position $i$ among token length $n$ in layer $l$, the top-k pivotal token method examines and sorts their variance.

$$\overline{S}_l = \{Y_{l,i} | i \in \text{top-k}(Var(Y_{l,i}))\}, i \in [0, ...n-1] \tag{8}$$

In this case, tokens with higher variance, namely outliers, are deemed pivotal tokens for each layer $l$. The Top-K pivotal tokens are thus selected for later MLP fusion in the LSTM module, as shown in Eq.8, this method is annotated as **VARIANCE (VAR)**. Furthermore, inspired by the findings that similar pivotal token patterns exist across different layers, the designed layer-wise aligned tokens' method applies a WTA-based method to select the same $k$ token positions for all layers, as shown in Alg. 2, namely **ALIGNED-VARIANCE (A-VAR)**. Rather than considering each layer's results independently, tokens are ranked based on their frequency of appearance as 'pivotal' across all layers. Those consistently identified are selected at last.

**LSTM-based Contribution Analysis** As depicted in Fig. 3, the classifier explores the layer-wise relationship between extracted MHA outputs. For each time step, LSTM cells are fed with outputs from sequential layers, where the MLP fusion module learns horizontal token aggregation, and then propagates layer-wise. The LSTM's final output is subsequently relayed to a 2-layer classifier, culminating in a softmax distribution that represents the contribution score of each potential class. During training, original copyrighted sub-datasets are fed into LLM, where the LSTM framework learns layer-wise dependency. In the inference stage, given specific user instructions, different responses generated by LLM are first fed into LLM itself for prefilling. This process is significant because feeding responses back into the original LLM triggers distinct patterns that are not observed

when using unrelated LLMs solely as feature extractors. Moreover, this approach maintains efficiency by relying only on prefilling. As such, semantic and causal information from both response and model are extracted by intermediate layer attention outputs. Such outputs are subsequently fed into *CopyLens* for copyright contribution analysis. The above two pivotal methods show different granularity, with the first offering specific understandings and the second ensuring consistency and robustness.

### 3.3 STAGE 1: CONTRASTIVE-LEARNING BASED NON-COPYRIGHT DETECTOR

To determine whether LLM outputs contain copyrights, we employ self-supervised learning exclusively on copyrighted datasets, as non-copyrighted outputs are uncontrollable. In this contrastive learning framework, all copyrighted datasets are treated as a single in-distribution (ID) class.

**Data Preparations and Training Settings** We aim to detect non-copyright datasets from $D$ directly. Copyright datasets $D_{\text{ID}}$ are first batched into $\{T_b\}_{b=1}^{|B|}$, and then augmented into $\{\tilde{T}_b\}_{b=1}^{|B|}$. We use the classic method Random Span Masking (RSM) for data augmentation. (Liu et al. (2021); Cho et al. (2022)) These data pairs are fed into a frozen LLM, generating layer-wise MHA outputs. They are extracted and processed by the LSTM contribution analysis module. The first-layer MLP output in this module is sent to the global projector, a lightweight MLP, producing two embeddings $e_b, \tilde{e}_b$. Finetuning the pre-trained LSTM and global projector with contrastive loss, we train an embedding space for non-copyright detection. The loss for each query text index $i$ is computed as shown in Eq. 9, with $\tau$ as the temperature hyperparameter.

$$L_{\text{CL}} = -\sum_{i=1}^{|B|} \log \left( \sum_{j=1}^{|B|} \frac{\exp\left(e_i \cdot \tilde{e}_j / \tau\right)}{\sum_{k=1, k \neq i}^{|B|} \exp\left(e_i \cdot \tilde{e}_k / \tau\right)} \right) \tag{9}$$

**Non-copyright OOD Detection** For OOD detection, we use Mahalanobis distance, a classic distance-based scoring function $D_M$ (Lee et al. (2018)), which measures the distance between a vector point $\mathbf{x}$ and a distribution $\mathbf{Q} \in \mathbb{R}^N$, as shown in Eq. 10. $\boldsymbol{\mu}$ and $\boldsymbol{\Sigma}$ represent mean and positive semi-definite covariance matrix in $\mathbf{Q}$, and $M$ is the dimensionality of $\boldsymbol{\mu}$. As in Eq. 4, the anomaly threshold $\delta$ decides whether a text is non-copyright, with the threshold set to maintain a 95% true positive rate (TPR) on the training samples.

$$D_M(\mathbf{x}, \mathbf{Q}) = \sum_{i=1}^{M} \left(\mathbf{x_i} - \boldsymbol{\mu}\right) \boldsymbol{\Sigma}^{-1} \left(\mathbf{x_i} - \boldsymbol{\mu}\right)^T \tag{10}$$

### 3.4 CONTROLLABLE TEXT GENERATION DESIGN

Since copyrighted dataset contribution has not been studied before, limited effort has been made to evaluate stage 2 due to uncontrollable user input. We consider three common controllable generation scenes, alongside prompt design and the text mixture task to quantitatively assess *CopyLens*.

**Continuation, Style Transfer, Style Elimination and Text Mixture** Without the advantages of flexibility in chat models, both prompt and generation scene design are essential for effectively utilizing vanilla language models. Therefore, we consider three controllable text generation types: continuation, style-transfer, and style-elimination, shown in Fig. 4. We carefully design prompts to ensure the response's validity.

Friendly to vanilla language models, text can be continued without additional prompts. For style-transfer instructions that users might give to rewrite inputs, we direct the style to concrete datasets, so that the transferring effects can be verified. In the style-elimination scene, prompts are designed to ask LLM to eliminate any possible styles. Detailed prompt designs are discussed in Appx.A.6.

To further evaluate *CopyLens* quantitatively, text mixture is designed in stage 2, where input texts into the framework are a mixture of multiple copyrighted training datasets with predefined portions.

## 4 EXPERIMENT

**Stage 1: Copyright Detection Task and Methods** The task is to decide whether a text is copyrighted, formulated as an OOD problem. While our main innovation lies in integrating two stages, ensuring

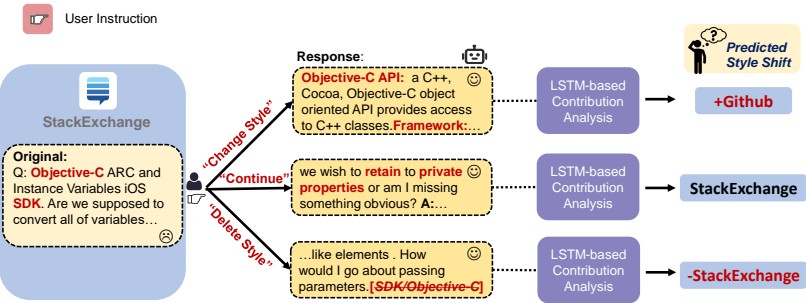

Figure 4: Overview of three controlled generation scenes: text continuation, style transfer, and style elimination. Shifts in dataset contributions from user editing instructions are expected to be detected.

reasonable accuracy in stage 1 is necessary. In stage 1, we propose a contrastive learning non-copyright detector with LSTM prior, and use (Cho et al. (2022)) as baseline for comparison.

**Stage 2: Contribution Analysis Tasks and Methods** We first construct a simplified classification task with copyrighted dataset classes as labels. Original copyrighted datasets are used as inputs rather than LLM outputs, and the collected MHA is divided into both train and test sets. Thus, in the inference stage, higher classification accuracy indicates better contribution analysis ability. After training on the classification task, our framework inferences on more realistic scenes. We consider three controlled generation scenes, continuation, style transfer, and style elimination. Additionally, we propose a task to detect text mixture ratios. The framework is trained on the original scenes and then tested on the others. It is expected to identify contribution shifts in the three generation scenes and accurately predict the mixture ratios in the text mixture task.

Three information extraction methods, INTER, VAR, and A-VAR, are applied in stage 2. Two baselines are proposed for the classification task. The prompt engineering baseline ($Baseline_1$/B1) is designed based on templates in Jiang et al. (2024). We treat LLMs as agents, asking LLMs to give out decisions according to given responses. Details are shown in Appx. A.6.To show the importance of layer-wise dependency, we build a distance-based method as another baseline ($Baseline_2$/B2), details are shown in Alg. 1.

**Models and Datasets** We test our methods over three open-source language models, including encoder-based and decoder-based ones: bert-base-uncased (Devlin et al. (2019)), GPT-Neo-2.7B (Black et al. (2021)) and GPT-J-6B (Wang & Komatsuzaki (2021)).

For the encoder-only model, we evaluate six GLUE classification sub-datasets: CoLA, SST-2, MRPC, WNLI, QQP, and RTE, with each sub-dataset comprising 500 training and 100 testing samples. Before further experiments, BERT is fine-tuned on tasks in GLUE. For the decoder-based models, experiments utilize the Pile dataset, an 825 GB open-source collection of 22 diverse sub-datasets. 8 classes including Github, OpenWebText2, Wikipedia (en), StackExchange, PubMed Abstracts, Pile-CC, USPTO Backgrounds, and FreeLaw are considered copyrighted datasets, and 4 classes including PubMed Central, Enron Emails, OpenSubtitles, and DM Mathematics are considered non-copyrighted ones.

## 4.1 RESULTS AND ANALYSIS

### 4.1.1 STAGE 1: NON-COPYRIGHTED DATASETS DETECTION

To compare with baseline Cho et al. (2022), 800 copyrighted data samples are used for self-supervised contrastive learning. 400 copyrighted and 200 non-copyrighted samples are used for the test set. Both methods use Random Span Masking (RSM) for data augmentation to ensure fairness. The OOD threshold is set to a 95% true positive rate (TPR). Despite using BERT backbone for fine-tuning, which takes up large computational resources, the baseline falls behind our proposed non-copyright detector, as depicted in Tab. 1. INTER, VAR and A-VAR shows consistent advantages.

Table 1: Non-copyright Dataset Detection Accuracy and AUC Comparison.

| OOD Detection | Inter. | | Var. | | A-Var. | | Baseline |
|---|---|---|---|---|---|---|---|
| | 2.7B | 6B | 2.7B | 6B | 2.7B | 6B | |
| Accuracy (%)(↑) | 92.97 | 90.80 | 88.95 | 93.83 | 83.92 | 88.33 | 66.37 |
| AUC (↑) | **0.954** | 0.934 | 0.917 | 0.953 | 0.886 | 0.935 | 0.742 |

### 4.1.2 STAGE 2: CLASSIFICATION PERFORMANCE AND DATASET CORRELATIONS

**Classification Performance Comparison** In classification task, $Baseline_1$, $Baseline_2$ and *Copy-Lens* are tested, as shown in Fig. 5. We allocate up to 12,800 training and 6,400 testing samples per copyrighted class. All methods are evaluated on the same, maximum test set. For the INTERVAL, VAR, and A-VAR methods, *CopyLens* achieves contribution analysis accuracies of 95.4%, 94.3%, and 94.9% respectively on GPT-Neo-2.7B, surpassing prompt engineering ($Baseline_1$) by 81.4%, 80.3%, and 80.9%, and our developed best baselines ($Baseline_2$) by 52.6%, 51.5% and 52.1%. Similar results are observed in GPT-J-6B.

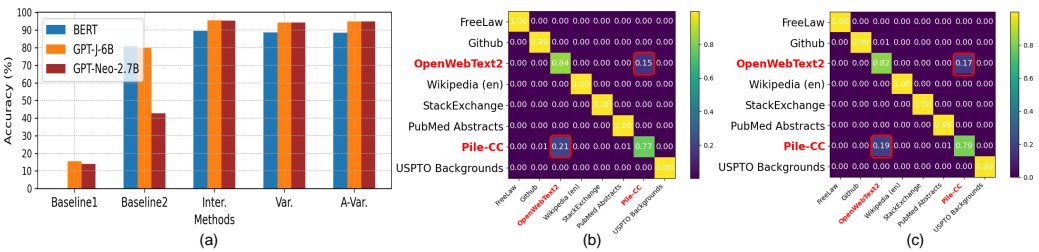

Figure 5: Classification performance comparison across various methods and language models. (a) Original classification scene performance comparison. (b,c) Self-correlation analysis for each sub-dataset using trained GPT-Neo-2.7B and GPT-J-6B analysis framework.

Despite $Baseline_2$ achieving over 80% accuracy in BERT, its performance drops drastically to 40% in GPT-Neo-2.7B and struggles below 80% in GPT-J-6B, even with training samples scaling up to 1,600. In contrast, *CopyLens* shows steady accuracy above 90% in each model.

**Self-correlation Predictions in Datasets** Correlations naturally occur between sub-datasets. To explore these relationships, we calculate the average probability distributions for each using *CopyLens* during inference with both GPT-Neo-2.7B and GPT-J-6B models. As shown in Fig. 5(b)(c), the Pile dataset, which includes OpenWebText and Pile-CC (both web-sourced), validates the effectiveness of our framework. Both models exhibit similar correlation patterns, reinforcing our approach.

### 4.1.3 STAGE 2: EVALUATION ON CONTROLLED GENERATION SCENES

In this task, *CopyLens* is tested on realistic LLM outputs. We design prompts to induce generated texts with "controlled" copyrighted elements. All related elements are transferred to "Github" style in style transfer, and removed in style elimination. Fig. 6 (a) shows that our designed prompt and framework are both valid, detecting drops from original to transfer, and slightly back on elimination, which aligns with the trend in baseline reference in Fig. 6 (b). The generation token length is set to 512, and experiments on varying lengths have been conducted to verify robustness in Tab. 4.

Both accuracy shifts from continuation to transfer are detected, with a much higher shift detection rate in *CopyLens*, as shown in Fig. 6(c). A-VAR outperforms INTER and VAR, consistent with the WTA mechanism, demonstrating robustness and generalization. In the style-transfer case, all texts are converted to "Github", and the contribution probability shift is measured between continuation and style-transfer. As shown in Fig. 6(d), INTER achieves the best detection rate, with over 20% change.

In conclusion, INTER outperforms variance-based methods, as semi-quantitatively explained with information theory proposed by us in Appx. A.8. However, while the mechanisms behind the effectiveness of outliers are not fully understood, variance-based methods significantly improve accuracy in larger models, whereas INTER shows minimal improvements, as shown in Fig. 6(c).

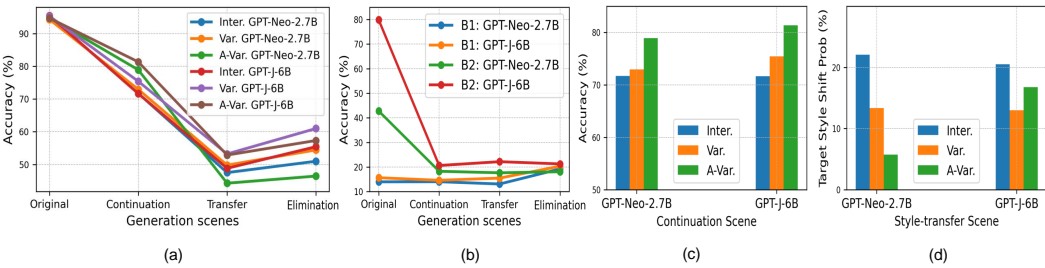

(a)            (b)            (c)            (d)

Figure 6: Contribution accuracy shifts comparisons of copyright detection methods in different generation scenes. Accuracy comparisons of proposed methods(a) and two baselines(b) in four designed scenes. (c) Accuracy comparisons of proposed methods in continuation scenes (d) Comparison of proposed methods in accuracy shift from continuation to transferred-style scene.

All methods in *CopyLens* can effectively detect mixture ratios with minimal loss. We randomly select 3 classes from 8 copyrighted datasets, and mix them into new samples by taking 15%, 15%, and 70% of the token lengths respectively. As shown in Fig. 7, predicted contribution from INTER, VAR and A-VAR aligns closely with ground truth ratios. More results are in Appx. A.7.

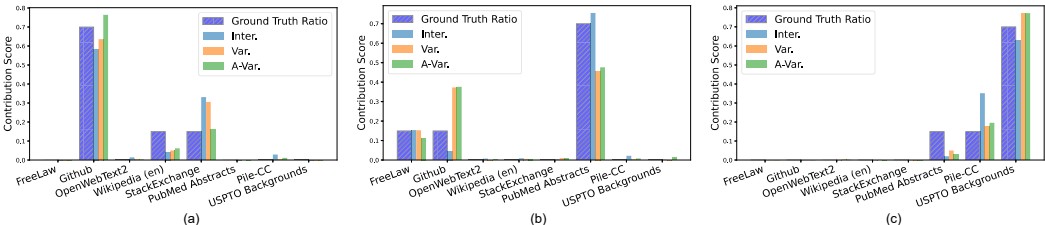

(a)            (b)            (c)

Figure 7: Comparison of predicted copyright contribution scores with ground truth ratios in text mixture task. Three random mixture examples are shown to illustrate effectiveness.

### 4.1.4 TRAINING EFFICIENCY COMPARISON AND DATA SCALE EFFECT

**Robustness to Data Scale** Despite copyright violations in limited fields, large amounts of copyrighted sub-dataset pose challenges to model training. Thus, a lightweight method with minimal data is needed. To test the robustness of data scaling in stage 2 in *CopyLens*, we separately limit training dataset samples to 160, 1,600, and 12,800, then inference with 6,400 samples. Besides the training data scale, the extensibility of sampled token number $k$ in the information extraction stage is also looked into, shown in Tab. 2. We find that training on 1,600 samples in sub-datasets is enough for *CopyLens* to outperform the best performance of baselines, further decreasing training time in Tab. 3. What's more, accuracy increases linearly with data scale and $k$. An empirical study is also performed, and a small number of $k$ is enough, in which case $k = 7$ shows model degradation.

**Training Efficiency** We conduct a comparative analysis of training time across different methods and models. As depicted in Tab. 3, $Baseline_2$ and *CopyLens* show superior performance in less than 1.5 hours, far more negligible than training or finetuning LLMs. (Thoppilan et al. (2022))

## 5 CONCLUSION

We highlight the need to identify which copyrighted datasets contribute most to LLM outputs. Therefore, the framework *CopyLens* is proposed from a model provider's perspective, exploring LLM architectures. Our two-stage framework extracts representations from MHA outputs, followed by LSTM-based analysis to identify copyrighted sub-datasets contributions. We further design a lightweight non-copyright detector based on LSTM priors, showing high accuracy and AUC. Validated across various real-world scenarios, including text continuation, style transfer, style elimination, and text mixture, *CopyLens* efficiently combines two stages for copyrighted data protection, detects dataset contribution with minimal data, and is robust to different generation lengths, outperforming prompt-engineering and distance-based baselines.

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

## A APPENDIX

### A.1 DETAILS OF DATA SCALE ROBUSTNESS EXPERIMENT

Table 2: Data scale effect across different methods and models. $k$ denotes the sampling token number. The model accuracy strictly follows the increase in $k$ and data scale until $k = 7$.

| Data scale (Accuracy%) | $k$ | Inter. | | Var. | | A-Var. | | Baseline1 | | Baseline2 | |
|---|---|---|---|---|---|---|---|---|---|---|---|
| | | **2.7B** | **6B** | **2.7B** | **6B** | **2.7B** | **6B** | **2.7B** | **6B** | **2.7B** | **6B** |
| Small (160,6400) | 1 | 69.6 | 75.6 | 63.4 | 75.7 | 75.4 | 70.8 | - | - | - | - |
| | 3 | 85.5 | 87.1 | 76.9 | 83.5 | 83.8 | 80.4 | - | - | - | - |
| | **5** | **85.9** | **88.6** | **79.7** | **85.9** | **85.9** | **82.7** | - | - | - | - |
| | 7 | 82.7 | 87.6 | 67.4 | 84.7 | 79.0 | 81.4 | - | - | 39.2 | 72.3 |
| Medium (1600,6400) | 1 | 88.6 | 89.0 | 89.9 | 91.2 | 89.5 | 88.3 | - | - | - | - |
| | 3 | 92.8 | 93.1 | 91.5 | 92.6 | 91.8 | 91.1 | - | - | - | - |
| | **5** | **93.3** | **93.6** | **92.3** | **93.2** | **93.1** | **91.3** | - | - | - | - |
| | 7 | 93.4 | 94.0 | 92.5 | 93.2 | 93.0 | 91.7 | - | - | 42.7 | 79.4 |
| Large (12800,6400) | 1 | 92.0 | 93.3 | 93.4 | 93.9 | 92.2 | 93.3 | - | - | - | - |
| | 3 | 94.8 | 95.4 | 94.3 | 95.3 | 94.3 | 94.4 | - | - | - | - |
| | **5** | **95.4** | **95.5** | **94.3** | **95.4** | **94.9** | **94.7** | - | - | - | - |
| | 7 | 95.2 | 95.7 | 94.8 | 95.3 | 95.2 | 95.1 | 14.0 | 15.6 | 42.8 | 79.9 |

### A.2 DETAILS OF DATASCALE EXPERIMENTS

Table 3: Method Efficiency Comparison

| Model | Vanilla Model Training | | *CopyLens* Training | |
|---|---|---|---|---|
| | **Time** | **Hardware** | **Time** | **Hardware** |
| BERT | 4 days | TPU $\times$ 16 | 17 mins Acc: 89% | A100-40G $\times$ 1 |
| GPT-Neo-2.7B | 90 days | A100-80G $\times$ 96 | 58 mins Acc: 95% | A100-40G $\times$ 1 |
| GPT-J-6B | 5 weeks | TPU $\times$ 256 | 89 mins Acc: 95% | A100-40G $\times$ 1 |

### A.3 EXTENDED RELATED WORK

**Data Filtering** Filtering out copyrighted data directly avoids copyright issues in LLMs. Previous studies show possible ways such as utilizing synthetic or deduplicated data Kandpal et al. (2022); He et al. (2023). Whereas, such approaches can detrimentally affect model performance due to the high quality of copyrighted datasets.

**LLM unlearning** Researches show that LLMs can memorize training datasets, which causes trials in letting LLMs forget training data. (Carlini et al. (2021); Eldan & Russinovich (2023)) However, the unlearning method was proved unreliable using a Min-k probability attack. (Shi et al. (2023))

**Watermark of copyrighted data** Adding a watermark to the copyrighted victim dataset makes tracing possible. Watermark based on linguistics features exploits statistical distribution, token entropy, and so on. (Wu et al. (2023); Kirchenbauer et al. (2023); Lee et al. (2023)) Despite being training-free, this method is vulnerable to attacks. (Panaitescu-Liess et al. (2024)) The token sampling method switches the watermark process to the LLM decoding stage, showing superior robustness. (Kuditipudi et al. (2023)) Similar approaches include adding a watermark during logit generation. (Liu et al. (2023)) End-to-end learning-based watermark methods are also proposed, which can also attribute data sources. However, the requirement of pre-training LLMs poses a significant computational burden. (Zhang et al. (2023); Wang et al. (2023a))

**Member Inference Attack (MIA)** Member Inference Attack aims at deciding whether specific data is included in training datasets. It usually requires a shadow reference model assuming this model shares similar training data distributions with the victim one. (Watson et al. (2021); Mattern et al. (2023)) However, it's hard to apply to black-box models and only serves the purpose of data provenance. (Shi et al. (2023))

### A.4 DETAILS OF THE $Baseline_2$

In this section, we explain the algorithm details of distance-based baseline. After the MHA outputs are collected, in each layer, the last token is selected for its substantial information because of the decoder-based autoregressive manner. The mean value is thus calculated across all layers to form a representation of a specific data sample. This algorithm works in a supervised way, where training samples in each sub-dataset are averaged into one high-dimensional vector. In the inference stage, L1 distance is calculated to decide which class the test sample should belong to, as depicted in Alg. 1.

---

**Algorithm 1** Distance-based Contribution Analysis ($Baseline_2$)

---

1: **Input:** Datasets, Language Model
2: **Output:** Class Identification Success Rates
3: Initialize dictionaries for layers
4: **for** $index$ in $len(Datasets)$ **do**
5:     **for** each $l$ in $layer$ **do**
6:         Compute the last token embedding across samples for each layer
7:     **end for**
8: **end for**
9: **for** each test sample $x_{test}$ in $TestSet$ **do**
10:     $d \leftarrow \|\mu_l - x_{test}\|_1$ for each layer $l$
11:     $c \leftarrow \arg\min_d \sum_l d_l^{(d)}$
12: **end for**
13: Calculate Total Accuracy
14: **return** Class Identification Success Rate

---

### A.5 ALGORITHM DETAILS OF PIVOTAL TOKEN INFORMATION EXTRACTION STRATEGY (A-VAR)

The algorithm details of the pivotal token information extraction strategy (A-VAR) are shown in Alg. 2. Fig. 8 shows the overview of two main information extraction strategies.

---

**Algorithm 2** Layer-wise Aligned Outlier Tokens Representation (A-VAR)

---

1: **Input:** MHA Outputs
2: **Output:** A Set of Aligned Token Positions
3: Initialize an empty set of aligned token positions $A_{index} \leftarrow \emptyset$
4: Initialize a dictionary for position occurrence counts $Num \leftarrow \{0, \ldots, n-1 \mapsto 0\}$
5: **for** each layer $l$ in $layers$ **do**
6:     Compute and select the top-$k$ token variance indices $S_l$
7:     **for** each index $i$ in $S_l$ **do**
8:         Increment $Num[i]$ by 1
9:     **end for**
10: **end for**
11: $A_{index} \leftarrow \text{top-}k(\{Num[i]\}_{i=0}^{n-1}, k)$
12: **return** $A_{index}$

---

### A.6 CONTROLLED TEXT GENERATION AND PROMPT ENGINEERING BASELINE EXAMPLES

To show the effectiveness of the designed prompt template, Fig. 9 presents an example of a style-transferring prompt. The figure shows how a text related to database security is accurately transformed into SQL code. An ablation experiment on generation length is shown in Tab. 4 to prove robustness.

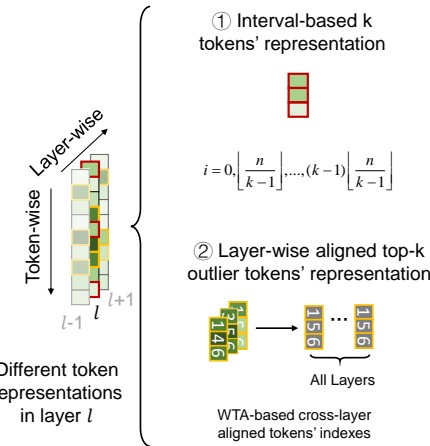

① Interval-based k tokens' representation

$$i = 0, \left\lfloor \frac{n}{k-1} \right\rfloor, ..., (k-1)\left\lfloor \frac{n}{k-1} \right\rfloor$$

② Layer-wise aligned top-k outlier tokens' representation

All Layers

WTA-based cross-layer aligned tokens' indexes

Figure 8: Annotations and explanation of the information extraction and token fusion process, showing two main methods.

| Continuation Scene | 64 | 128 | 256 | 512 |
|---|---|---|---|---|
| **Accuracy (%)** | | | | |
| INTER | 73.65 | 72.95 | 77.82 | 71.73 |
| A-VAR | 72.95 | 72.96 | 80.56 | 78.94 |
| VAR | 70.98 | 77.03 | 75.15 | 72.98 |

Table 4: Ablation experiments on robustness to different generation lengths on GPT-Neo-2.7B. The impact of text length on our proposed method is minimal, and our method is more suitable for real-world scenarios with longer texts.

Besides, the prompt engineering template for the baseline is provided. In this case, due to the lengthy dialog input, words such as "judge" and "charge" are ignored by the LLM. Instead, attention is given to the nearest word "search", leading to an incorrect answer, as illustrated in Fig. 10.

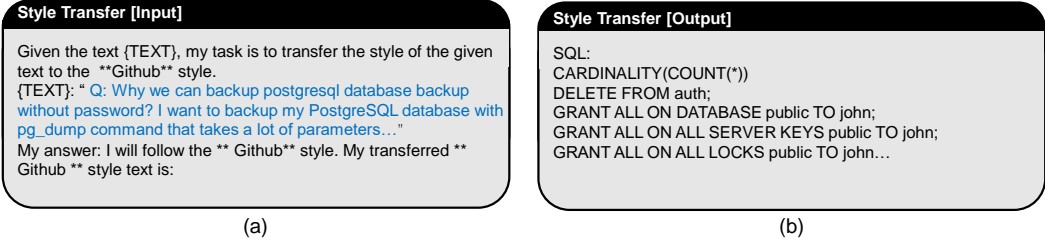

**Style Transfer [Input]**

Given the text {TEXT}, my task is to transfer the style of the given text to the **Github** style.
{TEXT}: " Q: Why we can backup postgresql database backup without password? I want to backup my PostgreSQL database with pg_dump command that takes a lot of parameters…"
My answer: I will follow the ** Github** style. My transferred ** Github ** style text is:

(a)

**Style Transfer [Output]**

SQL:
CARDINALITY(COUNT(*))
DELETE FROM auth;
GRANT ALL ON DATABASE public TO john;
GRANT ALL ON ALL SERVER KEYS public TO john;
GRANT ALL ON ALL LOCKS public TO john…

(b)

Figure 9: Prompt Design and LLM Output in Style Transfer Scene. (a) Designed prompt input. (b) LLM output.

**Prompt Engineering Attribution [Input]**

Given a text:
" { …We think the district judge sufficiently followed the approved Michelson charge so that the subordinate contention that the trial judge's instructions to the jury, even under Michelson, were insufficient… The search warrant contention is frivolous.}.
Which dataset you think the text belongs to ? You can choose from the following options: (A). FreeLaw (B). Github (C). OpenWebText2 (D). Wikipedia (en) (E). StackExchange (F). PubMed Abstracts (G). Pile-CC (H). USPTO Backgrounds.
My answer: I think the best Dataset the text belongs to is option (

(a)

**Prompt Engineering Attribution [Output]**

G).

There are many good reasons to choose this dataset, but one is that option (G) incorporates search terms that have been used to determine whether the entry belongs to any of the datasets already used on the Open Web. I feel this additional check is essential in order to eliminate duplication and to ensure that only unique entries are included in the text database.

(b)

Figure 10: Prompt Design and LLM Output in Prompt Engineering Baseline. (a) Designed prompt input. (b) LLM output.

## A.7 MSE LOSS IN TEXT MIXTURE TASK

In the text mixture task, eight copyrighted sub-dataset classes in the Pile $\{0, 1, 2, 3, 4, 5, 6, 7\}$ are used for mixture. We generate mixed text by sampling text segments based on predefined proportions (15%, 15%, 70%) for each label. The segments are then randomly combined to create the final mixed text. MSE loss is computed between predicted dataset contribution scores and predefined portions, which is lower than 0.05 in average, as Tab. 5 shows.

| MSE Loss($\downarrow$) | Mix Sequence | GPT-Neo-2.7B | | | GPT-J-6B | | |
|---|---|---|---|---|---|---|---|
| | (0.15, 0.15, 0.70) | INTER | VAR | A-VAR | INTER | VAR | A-VAR |
| | [0,7,6] | 0.00644 | 0.00245 | 0.00448 | 0.00925 | 0.00433 | 0.00184 |
| | [1,0,5] | 0.00229 | 0.01572 | 0.00501 | 0.00231 | 0.01814 | 0.01711 |
| | [1,4,3] | 0.00327 | 0.02104 | 0.00342 | 0.01119 | 0.04381 | 0.03847 |
| | [2,5,0] | 0.00773 | 0.00771 | 0.00729 | 0.00941 | 0.01251 | 0.00762 |
| | [3,1,4] | 0.03448 | 0.03803 | 0.01674 | 0.04416 | 0.03830 | 0.02203 |
| | [4,1,6] | 0.01747 | 0.02788 | 0.00652 | 0.00693 | 0.03489 | 0.03057 |
| | [4,7,0] | 0.00474 | 0.00295 | 0.00385 | 0.00410 | 0.00723 | 0.00559 |
| | [5,6,0] | 0.00440 | 0.00439 | 0.01920 | 0.02089 | 0.00949 | 0.00499 |
| | [7,4,6] | 0.00365 | 0.00887 | 0.00649 | 0.01149 | 0.00600 | 0.00336 |
| | Mean | 0.00939 | 0.01432 | 0.00811 | 0.01331 | 0.01941 | 0.01462 |

Table 5: MSE Loss for Different Models in Text Mixture Task.

## A.8 SIMPLE BUT EFFECTIVE: WHY DOES INTERVAL SAMPLING OUTPERFORM OUTLIER METHODS IN MOST CASES?

Tab. 2 shows that in vanilla classification scenes, interval-based sampling outperforms pivotal-token-based (outlier) methods. We try to explain why this happens from the perspective of information theory and take $k = 3$ in the smallest data scale as an example.

Why interval-based sampling method is effective? We formularize the overall information extraction and classification process based on mutual information (MI). MI quantifies the statistical association between variables, exploring both linear and non-linear dependencies, and can be used to identify relationships in potential causality. A larger MI shows a closer relationship between the two variables.

The process of the proposed framework can be summarized into two parts: first extracting token representations from MHA outputs, then feeding into the LSTM-based contribution analysis framework. The latter can be simplified as a classification. TODOTODOTODO need revise the framework

**Problem Formulation** We choose $k = 3$ with the smallest data scale as basic settings in Tab. 2. Therefore, in each layer $l$ with an input training sample $X$, given MHA outputs $O_l$ across all layers, the extracted three representations are denoted as $S_{1l}, S_{2l}, S_{3l}$ with the extraction policy $F_\theta$.

Our initial goal is to maximize the mutual information between extracted representations $S$ and the corresponding data sample $Y$ label. That is, $\max I(Y; S)$, where I(;) denotes mutual information.

Due to the constraints on the complexity of $S_l$, information must be efficiently compressed. To achieve this, we must minimize the MI between the MHA outputs and the extracted representations, as well as between the extracted representations themselves. Therefore, the final objective is to maximize the information bottleneck as shown in eq. equation 11:

$$\max_{F_\theta} \sum_l \left( I(Y; S_l) - \beta I(O_l; S_l) - \beta \sum_{1 \leq n1 < n2 \leq 3} I(S_{n1}; S_{n2}) \right) \tag{11}$$

where $\beta$ is set to 0.5, and

$$I(Y; S_l) = I(Y; S_{1l}) + I(Y; S_{2l}) + I(Y; S_{3l}) \tag{12}$$

$$I(O_l; S_l) = I(O_l; S_{1l}) + I(O_l; S_{2l}) + I(O_l; S_{3l}) \tag{13}$$

$$\sum_{1 \leq n1 < n2 \leq 3} I(S_{n1}; S_{n2}) = I(S_{1l}; S_{2l}) + I(S_{1l}; S_{3l}) + I(S_{2l}; S_{3l}) \tag{14}$$

We denote the above formulation as **Information Bottleneck (IB)**. The larger this metric is, the more necessary information the process maintains.

**Relationship between IB and Classification Accuracy** An intuitive idea is that better IB metrics mean higher accuracy. To investigate the relationship between the Information Bottleneck (IB) metric and classification accuracy, we conducted experiments on GPT-Neo-2.7B and GPT-J-6B. As depicted in Tab. 6 and Tab. 7, the IB metric closely correlates with classification accuracy. The interval-based method shows better IB metrics, which corresponds to its higher accuracy. This correlation may explain the superior performance observed with interval-based samples.

In summary, we have designed an IB-based metric that quantitatively explains the differences between extraction methods. This paves the way for research into optimized extraction strategies in theory.

| Methods | INTER. | VAR. | A-VAR. |
|---|---|---|---|
| Framework Classification Accuracy (%) | 85.5 | 76.9 | 83.8 |
| **Mutual Information** | | | |
| $I(O; S_1)(\downarrow)$ | -6.86 | 11.09 | 10.13 |
| $I(O; S_2)(\downarrow)$ | -1.63 | 12.63 | 12.46 |
| $I(O; S_3)(\downarrow)$ | 1.96 | 12.87 | 12.88 |
| $I(S_1; S_2)(\downarrow)$ | 1.52 | 42.87 | 28.79 |
| $I(S_1; S_3)(\downarrow)$ | 1.50 | 38.55 | 23.26 |
| $I(S_2; S_3)(\downarrow)$ | 18.04 | 42.68 | 27.59 |
| $I(Y; S_1)(\uparrow)$ | 36.44 | 1.17 | 2.78 |
| $I(Y; S_2)(\uparrow)$ | 0.17 | 0.39 | 0.63 |
| $I(Y; S_3)(\uparrow)$ | 1.44 | 0.31 | 0.65 |
| $I(O; S)(\downarrow)$ | -6.53 | 36.59 | 35.46 |
| **Final: Information Bottleneck($\uparrow$)** | 30.79 | -78.47 | -53.50 |

Table 6: MI comparison with interval-based and outlier methods on GPT-Neo-2.7B. The classification accuracy is from Tab. 2, and the calculated information bottleneck shows exactly the same trend with framework accuracy. Experiments are conducted under settings of $k = 3$ at the smallest data scale.

| Methods | INTER. | VAR. | A-VAR. |
|---|---|---|---|
| Framework Classification Accuracy (%) | 87.1 | 83.5 | 80.4 |
| **Mutual Information** | | | |
| $I(O; S_1)(\downarrow)$ | 0.95 | 3.97 | 12.75 |
| $I(O; S_2)(\downarrow)$ | 1.52 | 3.53 | 16.16 |
| $I(O; S_3)(\downarrow)$ | 1.54 | 3.25 | 17.11 |
| $I(S_1; S_2)(\downarrow)$ | 1.27 | 32.01 | 36.42 |
| $I(S_1; S_3)(\downarrow)$ | 1.18 | 28.60 | 33.02 |
| $I(S_2; S_3)(\downarrow)$ | 8.43 | 30.13 | 43.82 |
| $I(Y; S_1)(\uparrow)$ | 35.21 | 2.46 | 5.05 |
| $I(Y; S_2)(\uparrow)$ | 4.77 | 1.32 | 6.33 |
| $I(Y; S_3)(\uparrow)$ | 8.76 | 0.76 | 5.17 |
| $I(O; S)(\downarrow)$ | 4.01 | 10.75 | 46.01 |
| **Final: Information Bottleneck($\uparrow$)** | 41.29 | -46.21 | -63.09 |

Table 7: MI comparison with interval-based and outlier methods on GPT-J-6B. The classification accuracy is from Tab. 2, and the calculated information bottleneck shows exactly the same trend with framework accuracy. Experiments are conducted under settings of $k = 3$ at the smallest data scale.

