# OpenReview forum: "CopyLens: Dynamically Flagging Copyrighted Sub-Dataset Contributions to LLM Outputs"
_ICLR.cc/2025/Conference — ICLR 2025 Conference Withdrawn Submission_

### Official Review · Reviewer_rfNN · 2024-10-15

**Soundness:** 1
**Presentation:** 1
**Contribution:** 1
**Rating:** 3
**Confidence:** 4

**Summary:**

The authors introduce a method to analyze the contributions of copyrighted sub-datasets to language model outputs. The paper begins with a review of previous work, and proposes a two stage approach that first detects copyrighted material in a language model output, and then attributes that material to a sub-dataset in the training data. The authors propose using the multi head attention (MHA) states of a language model during the copyright detection step. The authors use a contrastive learning approach to take LSTM pooled MHA states and produce what they call an “in distribution” embedding. They then apply out of distribution detection methods to determine if a particular MHA activation is in or out of the copyright distribution.

**Strengths:**

The paper proposes an interesting method for out of distribution detection.

**Weaknesses:**

Figure 2 requires several modifications:

- A legend describing the meaning of the colors in the figure
- The authors describe the distribution of panel (c) as “filamentous.” The authors should be careful when drawing geometric conclusions from UMAP plots, since “filamentous” plots can result from UMAP parameters and not properties of the distribution itself.
- [257-259] The authors claim that this “filamentous” distribution contains “richer information” than  the untrained model, but provide no evidence for this claim.

At a higher level, the choice to use multi head attention parameters as the input to their copyright detection is somewhat curious, particularly since it is known in the mechanistic interpretability community that factual recall (and thus, potentially memorized content) is located in the feed forward blocks of transformers. (Meng et al 2022).

Moreover, the authors’ entire framework assumes that the MHA states of a model will discriminate between copyright and non-copyright partitions of the training data. Previous mechanistic interpretability work has shown that a large proportion of attention heads implement syntactic functions, including induction, copying, part of speech detection, etc. (Olsson et al 2022, Vig 2019), which may not be informative for detecting the copyrighted content present in a sequence.

While the authors’ present compelling results that their method indeed enables dataset classification based on MHA states, they do not provide evidence that this discriminability is based on copyrighted content, and not on differences in the syntax of datasets that may be activating structural attention heads at different rates.

The baseline comparisons for out of distribution (OOD) detection in the paper are weak. The only prior art that is compared to in Table 1 is Cho et al, and the performance of the baseline on the authors’ task is significantly below the OOD statistics reported in the original Cho et al paper (see e.g., Cho et al. table 2).

For the experiment present in Figure 6 authors use prompting to “induce generated texts with controlled copyrighted elements.” [472-475] However, the authors present no evidence that their prompt based techniques do indeed induce infringing model outputs. For instance, simply matching the “Github Style” [473] is not obviously sufficient for calling an output copyright protected.

Overall, the paper does not provide any evidence that their method can actually be to copyright detection. At best, the paper presents a method that classifies which dataset in a set of datasets that a particular model output is closest to based on MHA states.

**Questions:**

I would be interested in justification for the usage of multi head attention states as the input for the classifier, as well as any arguments that the proposed methods can be applied to the task of copyright detection.

---

### Official Review · Reviewer_MXQk · 2024-10-30

**Soundness:** 1
**Presentation:** 1
**Contribution:** 1
**Rating:** 1
**Confidence:** 3

**Summary:**

The paper proposes a framework called "CopyLens" to dynamically assess and flag contributions from copyrighted sub-datasets in LLM outputs. The study addresses the challenge of copyright protection in LLM pre-training data, where it is unclear how specific datasets influence generated responses. CopyLens uses a two-stage approach: the first stage detects whether outputs contain copyrighted content, and the second stage quantifies the contributions of training datasets. By processing Multi-Head Attention (MHA) outputs with LSTM and contrastive learning, CopyLens detect copyrighted outputs and the contributions of the training datasets. Experiments demonstrate that CopyLens outperforms baseline models in both efficiency and accuracy.

**Strengths:**

This study divided the copyright detection problem into two stages: detecting whether LLM outputs contain copyrighted content and quantifying the contributions of training datasets to the outputs.

**Weaknesses:**

- The problem setup in this study is merely a text classification task labeled as "copyright detection," without introducing a genuinely new problem. In the experiment described in Section 4.1.1 and 4.1.2, the study treats texts from sources like Github, OpenWebText2, and Wikipedia as "copyrighted," and texts from sources like PubMed and Enron Emails as "non-copyrighted". This problem setup is just a standard text classification task, and they have been extensively explored in numerous previous studies. Section 4.1.3 can also be viewed as a classification task on noisy text (e.g., text with different styles) rather than a real-world problem setting. It does not address the actual challenge of detecting copyright-infringing content within LLM outputs.


- The authors use the prompting method and their original method as baselines. However, there are many methods that perform well on text classification tasks. Also, while there are several works on copyrighted content detection tasks recently, they do not employ these works because they are "preliminary and not universally applicable", without mentioning their details.
> How does the inclusion of copyrighted material as training data in open-source models affect the resulting outputs? Mathematical
approaches like K-Near Access-Free (K-NAF) similarity have been developed, but these methods are preliminary and not universally applicable (Vyas et al. (2023); Scheffler et al. (2022); Elkin-Koren et al. (2023)).


- The text contains many points that I find difficult to understand. For example, in the following text, it is unclear what distinguishes the "original" version from the "simplified" version, as these are not defined. I also do not understand what data and labels are used for supervised training, nor what the identity matrix represents. Additionally, Equation (5) appears to maximize the probability of the function
𝐹, which is confusing to me.
> l210-214: To optimize Fθopt in this problem, we first reduce it to a simplified version, and then generalize it back to the original one. In the training stage, for each potentially copyrighted sub-dataset k, we optimize mapping strategy F using the original datasets as input in a supervised way, as shown in Eq. 5. In ∈ Rn×n denotes the identity matrix since inputs are supervised, and we denote optimized parameters as θsim:

**Questions:**

What is the rationale behind this complex method? Why are many components, such as MHA outputs and LSTMs, needed for classification? Did you try more simple methods (e.g., LLM + adapter)?

---

### Official Review · Reviewer_94zM · 2024-11-03

**Soundness:** 1
**Presentation:** 1
**Contribution:** 1
**Rating:** 3
**Confidence:** 3

**Summary:**

The paper attempts to (a) detect which copyrighted datasets were a part of the training, and (b) compute the contribution of datasets in LLM outputs. This is useful to protect copyrighted information or incentivize (through payments) dataset providers/curators. The paper attacks this problem through a two stage approach wherein they first "filter out" non-copyrighted LLM outputs, and then analyzes how the copyrighted datasets contribute towards a given response.

**Strengths:**

I believe the broader goal of characterizing the value of a (possibly copyrighted) dataset is super important. Computing the value a dataset presents for a given response opens up ways for revenue sharing, or other incentive mechanisms to support dataset/content providers. This may ultimately lead to a healthier ecosystem, wherein creators are valued for their contributions and model developers in turn benefit from higher quality data.

**Weaknesses:**

There are many choices that are hard to motivate: for instance, the task of detecting whether a model generated a copyrighted response is modeled as a supervised learning problem, assuming full access to the model and its training dataset where it is a-priori known which training subsets are copyrighted and which are not. I believe these assumptions are highly limiting and do not reflect the current scenario. There is a veritable subfield of "dataset inference" (which this paper largely ignores) where the assumptions are much more milder, and the goal is to identify whether a black-box model was trained on a given dataset or not, assuming no knowledge of training data. (I understand that dataset inference solves a slightly different problem from the one in the paper, but there is plenty to learn from that setup, especially in terms assumptions).

Additionally, if the models are trained on *both* copyrighted and non-copyrighted subsets, how is the former considered "in-distribution" and the latter "out-of-distribution"? Aren't both a part of the training distribution, which "distribution" is implied here? Similarly, the paper formulates the task of detecting copyrighted content in LLM output as the task of "OOD detection," which does not make sense.


The writing of the paper is *extremely challenging to parse*. Often it requires reading a sentence multiple times and re-reading the paragraphs just to guess what the authors might have to say. Examples abound: take the third line of the abstract for instance, "Previous methods either focus on the defense of identical copyrighted outputs or find interpretability by individual tokens with computational burdens". (The sentences before or after do not elucidate what authors mean by "interpretability by individual tokens" and what it has to do with copyright issues). Just by reading the abstract and the introduction, it is hard to note what is the broad sketch of the solution, what is the nature of contribution. My concerns about writing overlook formatting issues (e.g., citations do not come after periods of sentences they are intended for).

**Questions:**

If the models are trained on a set of copyrighted and non-copyrighted datasets, how is the former considered "in-distribution" and the latter "out-of-distribution"? What does distribution mean here?

---

### Note · Authors · 2025-01-01

I have read and agree with the venue's withdrawal policy on behalf of myself and my co-authors.